

# Statistics of Green's functions on a disordered Cayley tree and the validity of forward scattering approximation

Pavel A. Nosov[1], Ivan M. Khaymovich[2,3*],
Andrey Kudlis[4] and Vladimir E. Kravtsov[5,6]

**1** Stanford Institute for Theoretical Physics, Stanford University,
Stanford, California 94305, USA
**2** Max-Planck-Institut für Physik komplexer Systeme,
Nöthnitzer Straße 38, 01187-Dresden, Germany
**3** Institute for Physics of Microstructures, Russian Academy of Sciences,
603950 Nizhny Novgorod, GSP-105, Russia
**4** Department of Physics and Engineering, ITMO University, St. Petersburg, 197101, Russia
**5** Abdus Salam International Center for Theoretical Physics,
Strada Costiera 11, 34151 Trieste, Italy
**6** L. D. Landau Institute for Theoretical Physics, Chernogolovka, Russia

⋆ ivan.khaymovich@gmail.com

## Abstract

The accuracy of the forward scattering approximation for two-point Green's functions of the Anderson localization model on the Cayley tree is studied. A relationship between the moments of the Green's function and the largest eigenvalue of the linearized transfer-matrix equation is proved in the framework of the supersymmetric functional-integral method. The new large-disorder approximation for this eigenvalue is derived and its accuracy is established. Using this approximation the probability distribution of the two-point Green's function is found and compared with that in the forward scattering approximation (FSA). It is shown that FSA overestimates the role of resonances and thus the probability for the Green's function to be significantly larger than its typical value. The error of FSA increases with increasing the distance between points in a two-point Green's function.

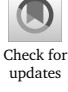

## Contents



# 1  Introduction

The forward-scattering approximation (FSA) for disordered quantum systems is the simplest approximation for describing the Anderson and many-body localization at strong disorder [1–5], which for some situations [6,7] is well corroborated by numerics. It takes into account only the *non-repeating paths* connecting two points $i$ and $j$ in the Green's function $G_{ij}(E) = (E - \widehat{H})^{-1}_{ij}$, and only those with the shortest length ("spaths") equal to the distance $r_{ij}$:

$$G_{ij}(E) = \sum_{spaths} \prod_{p \in spath} \frac{V}{E - \varepsilon_p}, \tag{1}$$

where $\varepsilon_p \in [-W/2, W/2]$ is the box-distributed random on-site energy, and $V = 1$ is the nearest-neighbor hopping amplitude. The distance $r_{ij}$ is defined on the corresponding graph or lattice by the minimal number of edges needed to hop in order to connect points $i$ and $j$.

The simplest situation is when there is only one path between the points $i$ and $j$, as it happens in one-dimensional systems and on the Cayley tree. In this case $\ln|G_{ij}(E)| \equiv \ln|G|$ is a sum of $r_{ij}$ i.i.d. random variables $\ln|V/(E - \varepsilon_p)|$, and for the box-shaped distribution of $\varepsilon_p$ and at $E = 0$ the PDF $P(\ln|G|)$ is the Poisson distribution, shifted by $r \ln(W/2)$ [4]:

$$P_{\text{FSA}}(z = \ln|G| + r \ln(W/2)) = \frac{z^{r-1}}{(r-1)!} e^{-z}. \tag{2}$$

The condition of non-repeating paths that does not pass through the same point twice, even along the same set of links, usually applies to strong disorder $W \gg V$. The naive reason for this condition is that increasing the length of the path by an extra link brings about a small factor $V/W \ll 1$. Such a justification, however, ignores completely the possibility of resonances when $|E - \varepsilon_i| < V \ll W$. This makes the status of FSA uncertain even at large $W/V$, especially on the "loopy" lattices or graphs like a hypercube lattice of Quantum Ising model [8–11] and Quantum Random Energy model [12–14] or its cross-section of XXZ Heisenberg chain [15–21]. In such lattices there are many paths of the same length $r$ which interfere with each other.

The necessity to evaluate the random Green's functions $G_{ij}(E)$ and their distribution function emerges in many problems, of which probably the first was the problem of mesoscopic fluctuations and magneto-resistance in strongly disordered semiconductors [22]. The present interest to FSA is boosted by the study of many-body localized states of disordered interacting systems [2–5].

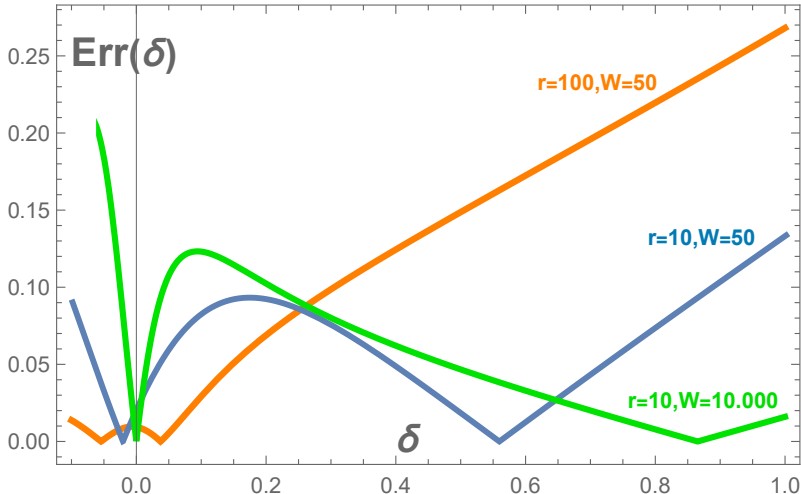

Figure 1: **Quality of FSA** The relative error $\mathrm{Err}(\delta) = |[\ln P(\ln|G|) - \ln P_{FSA}(\ln|G|)]/\ln P_{FSA}(\ln|G|)|$ in $\ln P(y = \ln|G|)$ as a function of the relative deviation $\delta = \ln(|G/G_{typ})/|\ln G_{typ}|$ from the typical value $G_{typ}$ for $r = 10$ and $r = 100$ at modestly strong disorder $W = 50$ and for $r = 10$ at extremely strong disorder $W = 10.000$. In all the cases the typical value $G_{typ}$ and the distribution function of $\ln|G|$ at small deviations from it is well described by FSA, while at a sufficiently large deviation ($\delta > 0.05$ at $W = 50$, $r = 100$; $\delta > 0.55$ at $W = 50$ and $r = 10$; $\delta > 0.85$ at $W = 10.000$, $r = 10$ ) the error changes the sign (resulting in a cusp on a plot of its absolute value) and increases indefinitely as the deviation further grows. Extremely large disorder may delay the onset and reduce the slope of this growth but it does not suppress the error completely. At a large distance $r_{ij}$ between the points $i$ and $j$ in $G_{ij}(E)$ the overestimation of $\ln P(y = \ln|G|)$ by FSA at large $|G| > G_{typ}$ is greatly enhanced.

In this paper we show that even for the Anderson model on a Cayley tree where there is only one geometric path from the initial to the final point, the status of FSA is subtle. The typical value $G_{typ} = \exp[\langle\ln|G|\rangle]$ and the distribution function $P(\ln|G|)$ at small deviations from the typical value is described quite well by FSA in all the cases. However, FSA greatly overestimates the probability for sufficiently large deviation from the typical value at $|G| > G_{typ}$ (see Fig. 1), especially at a large distance $r$ between the initial and a final points in the Green's function. By increasing disorder one can suppress this error but for large $r$ it happens only at an unrealistically strong disorder.

The paper consists of three parts. In the first part we show, using the Efetov's super-symmetry method [23], that the moments $m = 2\beta < 1$ of the *real* Green's function $G_{ij}(E)$ on a Cayley tree are *exactly* expressed in terms of the largest eigenvalue $\epsilon_\beta$ of the *linearized* transfer-matrix (TM) equation [24] in the large $r_{ij} = r$ limit:

$$\langle|G_{ij}|^{2\beta}\rangle = c_\beta \left[\epsilon_\beta\right]^r. \tag{3}$$

This allows to compute the distribution function $P(y = \ln|G|)$ by the Mellin transform in the saddle-point approximation. In the second part, we derive approximate formulas for $\epsilon_\beta$ at large $W \gg 1$. Finally, we compute the distribution function $P(y = \ln|G|)$ at large $W \gg 1$ for different relations between $r$ and $W$ and discuss the accuracy of FSA.

Note that Eq. (3) is important in its own right. The point is that at small and intermediate disorder one has first to solve a non-linear integral equation in order to find a "renormalized" distribution of on-site energies and only afterwards to solve a linear spectral problem to obtain

$\epsilon_\beta$ for the renormalized distribution of disorder. This renormalization is highly non-trivial, and it is absent in the nonlinear sigma-model (NLSM) version of the problem [25]. The linear spectral problem emerges in the asymptotic regime of TM equation in which certain (Liouvillian) factor in the kernel is dropped. Likewise, Eq. (3) is also valid in the asymptotic regime $r \gg 1$, which, however, looks totally different from that in the TM equation. Yet, it is proven in this paper that it is the same function $\epsilon_\beta$ that controls both the dynamics of the kink solution of TM equation and the moments of Green's functions on a Cayley tree *at any disorder*.

If this non-trivial point is taken for granted, one may guess [26, 27] Eq. (3) from the results of Ref. [28] and Ref. [25]. The first of these works employed the one-step replica symmetry breaking (RSB), while the second one used the super-symmetric NLSM machinery. Despite difference in the methods and interpretation of the results, the basic equations in these two works appeared to be exactly the same. From the identity of these equations one may immediately deduce the close relationship, Eq. (3), between the moments of real Green's functions (entering the formalism of RSB) and the largest eigenvalue $\epsilon_\beta$ of the linearized TM equation (governing the dynamics of the kink solution in the framework of NLSM). In this work we present a formal derivation of Eq. (3) by the supersymmetry method but without the constraint of the NLSM which significantly simplifies the TM equation. Thus our paper proves the validity of Eq. (3) for a Cayley tree with one orbital per site rather than for an infinite number of orbitals per site as in NLSM. An alternative way of justifying Eq. (3) is presented in Ref. [29].

## 2   TM equation and renormalization of disorder distribution

The model Hamiltonian is given by

$$H = T + V, \quad V_{km} = \varepsilon_k \delta_{km}, \quad T_{mk} = T_{km}, \tag{4}$$

where $k, m = 1, .., N$, $N$ is the number of graph nodes, and $T$ is the symmetric dimensionless adjacency matrix describing a tree with $K + 1$ nearest neighbors in the bulk (and the root has only $K$ nearest neighbors). We also assume $T_{kk} = 0$. The on-site energies $\varepsilon_k$ are identically distributed according to the distribution function $F(\varepsilon)$.

The retarded (advanced) Green's functions $G_{nm}^\pm(E) \equiv (E - H \pm i\eta)_{nm}^{-1}$ can be represented via functional integral over commuting ($S_{R/A}$) and anti-commuting ($\chi_{R/A}$) variables as follows [23]:

$$G_{nl}^\pm(E) = -\int \prod_k [d\Phi_k d\Phi_k^\dagger] \, \chi_{R/A}(n) \chi_{R/A}^*(l) e^{-S_0[\Phi, \Phi^\dagger]},$$
$$S_0[\Phi, \Phi^\dagger] = -i \sum_{mk} \Phi_m^\dagger L \left(E\delta_{mk} - H_{mk} + i\eta\Lambda\delta_{mk}\right)\Phi_k, \tag{5}$$

where the super-vector $\Phi_k$ is defined as:

$$\Phi_k = (S_R(k), \, \chi_R(k), \, S_A(k), \, \chi_A(k))^T, \quad k = 1, ..., N \tag{6}$$

and the symmetry-breaking matrices $L = \text{diag}\{1, 1, -1, 1\}$ and $\Lambda = \text{diag}\{1, 1, -1, -1\}$. The matrices $\Lambda$ and $L$ are introduced to ensure correct analytic properties of the Green's functions (and convergence of the integrals). The measure is defined as

$$[d\Phi_k d\Phi_k^\dagger] \equiv -\frac{d^2 S_R(k) d^2 S_A(k)}{\pi^2} d\chi_R^*(k) d\chi_R(k) d\chi_A^*(k) d\chi_A(k). \tag{7}$$

Using this functional representation one can average over random $H_{nm}$ at an initial stage thus replacing the quadratic in $\Phi_k$ action $S_0$ by the non-quadratic one $S[\Phi, \Phi^\dagger]$ and then to evaluate

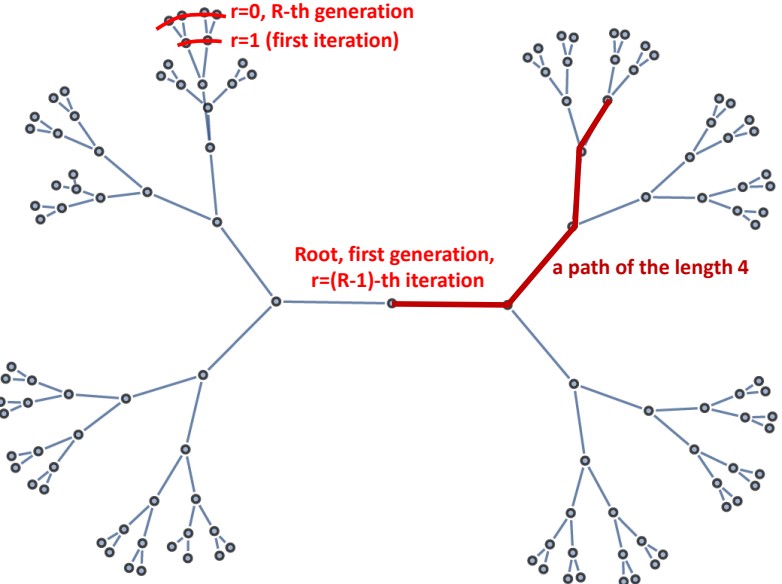

Figure 2: **Cayley tree with K=2.** Iterations start from the boundary ($R$th generation). The root (first generation) corresponds to $(R-1)$ iterations. A path between the root and a point at a distance 4 is also shown.

the generating functional:

$$Y(\Phi_n, \Phi_n^\dagger) = \int \prod_{k \neq n} [d\Phi_k d\Phi_k^\dagger] e^{-S[\Phi,\Phi^\dagger]}, \tag{8}$$

which allows to compute various observables. Remarkably, for the one-dimensional system ($K = 1$) and for the Cayley tree ($K > 1$) this calculation can be done by iterations starting from the boundary of a tree where $Y(\Phi_n, \Phi_n^\dagger) = 1$ and moving towards the root according to the transfer-matrix equation [24,30]. The superscript $r$ in this equation enumerates *iterations*, the number that enumerates *generations* on a tree is $R - r$, where $R$ is the total number of generations, see Fig. 2. The root corresponds to the first generation which is reached after $R - 1$ iterations. In enumerating the *nodes* of a tree it is natural to denote the root by $n = 1$.

For a Cayley tree with one orbital per site the generating function $\Omega_r(t, v)$ obeys the transfer-matrix equation:

$$\Omega_{r+1}(t, v) = \int_{-\infty}^{\infty} dt' dv' L(t - t', v, v') e^{-e^{t'}} [\Omega_r(t', v')]^K. \tag{9}$$

In Eq. (9) we denote $Y(\Phi_r, \Phi_r^\dagger) \equiv [\Omega_r(t, v)]^K$ with $e^t = \eta(|S_R|^2 + |S_A|^2)$ and $v = (|S_R|^2 - |S_A|^2)$, and

$$L(t, v, v') = \frac{1}{2\pi} e^{t/2} \cos\left(v' e^{t/2} + v e^{-t/2}\right) e^{iEv'} \tilde{F}(v'), \tag{10}$$

where $\tilde{F}(v)$ is the Fourier-transform (characteristic function) of the *bare* distribution of on-site energies.

The generating function $\Omega_r(t, v)$ serves to compute various observables at a point $r$ by integration over $t, v$ in the corresponding integral forms. The physical meaning of two variables $t$ and $v$ could be traced back to the imaginary and the real parts of the single-point Green's function [30]. The variable $t$ is indispensable to describe the blowing up of the typical imaginary part of Green's function in the delocalized phase as $r$ increases. It is just undefined if

the bare level width $\eta$ is set zero. On the other hand, $\eta$ should be smaller than the mean level spacing $\sim K^{-R}$ and should tend to zero before $R \to \infty$. The dependence of the generating function $\Omega_r(t, v)$ on the second variable $v$ describes details of the spectrum which encode residual effects of ballistic motion.

A similar equation can be derived [23, 25, 31] for a NLSM on a Cayley tree which corresponds to an infinite number of orbitals per site. However, in this case, the variables $v, v'$ do not appear in all the equations due to the NLSM constraint $Q^2 = 1$.

The next step is to consider the *self-consistent* solution $\Omega_r^{(sc)}(t, v) = \Omega_{sc}(t + ur, v)$ to Eq. (9) which the function $\Omega_r(t, v)$ tends to at $r$ sufficiently far from the boundary. At the boundary $\Omega_r(t, v) \equiv 1$ corresponds to the closed boundary conditions. In the delocalized phase it describes a kink in the variable $t$ running with a velocity $u$ as $r$ increases. In this kink, the solution $t$ plays a role of displacement, while $r$ plays a role of time. Far from the kink center, formally at $t \to -\infty$, the self-consistent variable takes the form:

$$\Omega_{sc}(t \to -\infty, v) = \Omega_0(v) + f_\beta(v) e^{\beta t}, \tag{11}$$

where the second term is a small perturbation. The function $\Omega_0(v)$ describes a non-trivial profile in the variable $v$ of the self-consistent solution at $t = -\infty$. As we will show in Sec. 4, physically, $\Omega_0(v)$ corresponds to a Fourier transform of the distribution function for real local Green's functions. It should be found from the solution of a *non-linear* equation:

$$\Omega_0(v) = \int_{-\infty}^{\infty} dv' \, \Xi_0(v, v') \, e^{iEv'} \, \tilde{F}(v') [\Omega_0(v')]^K \,, \tag{12}$$

with the kernel:

$$\Xi_\beta(v, v') = \frac{1}{\pi} \int_0^{+\infty} \frac{dz}{z^{2\beta}} \cos\left(v'z + vz^{-1}\right) . \tag{13}$$

Note that in the case of a *granular* Cayley tree described by a NLSM the profile of $\Omega_0(v) = 1$ is trivial. This is a significant simplification which stems from the delta-function distribution of the local Green's functions at an infinite number of states (orbitals) per granule.

Then linearizing TM Eq. (9) for $\Omega_r^{(sc)}(t, v)$ and omitting the Liouvillian factor $e^{-e^t}$ in the $t \to -\infty$ asymptotic regime one obtains the spectral problem:

$$\epsilon_\beta f_\beta(v) = \int_{-\infty}^{+\infty} dv' \, \Xi_\beta(v, v') \, \tilde{F}(v') \, e^{iEv'} [\Omega_0(v')]^{K-1} f_\beta(v') . \tag{14}$$

The solution to this spectral problem determines the velocity $u_\beta = \beta^{-1} \ln(K\epsilon_\beta)$ of the moving kink solution to the TM equation which minimization with respect to $\beta$ yields the fractal dimension of the wave function $D = \min_\beta[u_\beta]/\ln K$ determining its support set volume $\sim N^D$ [25, 28]. The localization transition corresponds to $\min_\beta[u_\beta] = 0$. Due to the symmetry $\epsilon_\beta = \epsilon_{1-\beta}$ the Anderson transition corresponds to the condition $\epsilon_{1/2} = 1/K$ at the symmetric point $\beta = 1/2$. The transition is controlled by the energy $E$ and the disorder strength $W$ entering the characteristic function $\tilde{F}(v)$ of the distribution of on-site energies and thereby in $\epsilon_\beta$.

The form of Eq. (14) immediately suggests the physical meaning of the factor $[\Omega_0(v)]^{K-1}$ as the factor that renormalizes the Fourier-transform of the on-site disorder distribution:

$$\tilde{\mathcal{F}}(v) = \tilde{F}(v) [\Omega_0(v)]^{K-1} . \tag{15}$$

On the other hand, as $\tilde{F}(v)$ is the generation function of the on-site disorder $\varepsilon_p$, this renormalization must take into account the self-energy parts of all the $K$ single-site Green's functions

linked to the current site, except the one which is on the considered TM path (see Abou-Chacra, Thouless, and Anderson equation from Ref. [24]). This tells us that $\Omega_0(\nu)$ should be closely related with the local Green's functions. Indeed, as it is shown in Sec. 4 it is the Fourier transform of the distribution function for each of these local Green's functions.

Eq. (15) shows that such a renormalization is absent in two special cases: (i) one-dimensional system $K = 1$, and (ii) the NLSM on the Cayley tree $\Omega_0(\nu) = 1$. In the second case the eigenvalue $\epsilon_\beta$ does not depend on the branching number $K$ of the tree.

It is instructive to show application of Eqs. (12), (15) to the exactly solvable case of the Cauchy distribution $\tilde{F}(\nu) = e^{-(W/2)|\nu|}$. At $\beta = 0$ Eq. (13) results in a singular kernel:

$$\Xi_0(\nu, \nu') = \delta(\nu') - \theta(\nu\nu') \sqrt{\left|\frac{\nu}{\nu'}\right|} J_1(2\sqrt{|\nu\nu'|}), \tag{16}$$

where $J_m(x)$ is the Bessel function of $m$th order and $\theta(x)$ is the Heaviside step function. One can easily find a solution to Eq. (12) in a form $\Omega_0 = e^{-\kappa|\nu|}$, with

$$\kappa = \frac{\sqrt{(W/2)^2 + 4K} - (W/2)}{2K}. \tag{17}$$

Then Eq. (15) gives rise to $\tilde{\mathcal{F}}(\nu) = e^{-(W_R/2)|\nu|}$ of the same Cauchy form with the renormalized "disorder strength":

$$W_R = \frac{1}{2K}\left((K+1)W + (K-1)\sqrt{W^2 + 16K}\right), \tag{18}$$

which at $K > 1$ remains finite as the "physical disorder" $W$ tends to zero.

The physical meaning of the renormalization, Eqs. (15), (18), is related to the remnant of ballistic motion in the disordered system. Formally, it arises because of the presence of $16K$ under the square root in Eq. (18). This effect is similar to the $K$-dependence of the spectral bandwidth due to kinetic energy contribution which remains finite as the disorder contribution to the bandwidth $= W/2$ tends to zero. Thus the "renormalization of disorder strength" is similar to the renormalization of the spectral bandwidth by kinetic energy contribution with respect to the width cased by pure on-site disorder. As it was mentioned above, this type of renormalization (but without keeping the same the functional form of the distribution $\tilde{\mathcal{F}}(\nu)$) happens for a generic $\tilde{F}(\nu)$ by a replacement $\tilde{F}(\nu) \to \tilde{\mathcal{F}}(\nu)$ according to Eq. (15).

It is especially important at small disorder $W \lesssim 1$. In particular this renormalization is responsible for a finite derivative $\partial_\beta \epsilon_\beta|_{\beta=0} = -\ln K$ which results in a finite Lyapunov exponent $\lambda_{typ} = -(1/2)\partial_\beta \epsilon_\beta|_{\beta=0}$ [28] at a vanishing disorder $W \to 0$ in the case of a Cayley tree with one orbital per node. The Lyapunov exponent describes the typical decrement of an exponentially decreasing spike of a wave function amplitude in a typical wave function. On a Cayley tree with one orbital per site the delocalized phase emerges as a proliferation of the *number* of isolated spikes with a finite decrement rather than by vanishing the decrement (Lyapunov exponent) of a one single spike. The Lyapunov exponent remains finite all the way down to vanishing disorder where it hits the ergodic limit value $\lambda_{typ} = (1/2)\ln K$. In the case of NLSM on a Cayley tree, where the renormalization is absent, the Lyapunov exponent tends to zero in the limit of vanishing disorder, as well as in a one-dimensional system. The ergodic limit in this case is reached at a finite disorder strength. This peculiarity of NLSM on a Cayley tree leads to an existence of an ergodic phase at a finite disorder strength [25], while the states are non-ergodic down to $W = 0$ on a disordered Cayley tree with one orbital per site [28, 29]. This difference between NLSM and the Anderson model with one orbital per site on the Cayley tree does not affect the scaling properties of transition between the non-ergodic extended and the localized phases.

# 3 Moments of real Green's functions

The goal of this section is to prove the relation, Eq. (3), between the moments of real Green's functions $I_\beta$ and the maximal eigenvalue of the linearized TM equation. To achieve this goal we make use of the Efetov's super-symmetry approach [32] but without passing to the sigma-model which is not justified for models with one orbital per site. This approach is essentially similar to the one of Ref. [33]. However, the concrete application of the super-symmetry technique to our problem is completely new in all its aspects and the results obtained.

By means of exact integration over the relevant variables of a super-vector $\Phi$ we obtain the closed expression, Eq. (48), for the moments of a two-point Green's function. It has a form of a multiple integral over the variables $v_k$ in the sites $k$ both along and beyond the path $\mathcal{P}$ connecting the two points in the Green's function. Finally, this multiple integral is represented in the iterative way similar to the TM equation (9). Importantly, integration over the variables $v_k$ in the sites belonging to $\mathcal{P}$ is described by the same spectral problem as Eq. (14), which kernel depends on the renormalized disorder distribution function. This renormalization appears to be identical to the one given by the function $\Omega_0(v)$ in Eq. (15), and the function $\Omega_0(v)$ itself emerges as the result of integration over the variables $v_k$ in the sites beyond the path $\mathcal{P}$. So we were able to proof a connection between two seemingly different problems: the spectral problem of the linearized TM equation and the problem of calculating the moments of a real two-point Green's function.

Below we present the details of our proof step by step. The readers who are not interested in the details of this calculation may go straight to Section 4 where we reveal the physical meaning of the renormalization factor $\Omega_0(v)$ in Eq. (15).

The focus of our interest in this paper is the moment $I_\beta$ the absolute value of a real Green's function $G_{nm}(E)$:

$$I_\beta = \left\langle |G_{nm}(E)|^{2\beta} \right\rangle, \quad 0 < \beta < 1/2. \tag{19}$$

In the absence of degeneracy of spectrum, Green's functions have simple poles at any $E = E_n$, where $E_n$ is an eigenenergy in a finite system. Upon averaging over disorder at a fixed $E$ the random levels hit in some disorder realizations the energy $E$ and cause very large values of $|G_{nm}(E)|$. This gives rise to a power-law tail of the distribution function $P(x = |G|) \sim \langle \delta(x - |G_{nm}(E)|) \rangle \approx \int dE_k \, \delta(x - |E - E_k|^{-1}) \sim 1/G^2$. Such a tail makes the moments of real Green's functions of order $2\beta \geq 1$ divergent, while this is not the case for complex Green's functions with $\eta$ greater than the mean level spacing $\delta \sim K^{-R}$. At the same time the moments of real Green's functions of order $2\beta < 1$ are well-defined. In this case the limits $\eta \to 0$ and $R \to \infty$ commute, and the moments of *real* Green's functions at a finite system size are given by the limit $\eta \to 0$ of the corresponding moment of $G^+ G^-$ involving complex Green's functions $G_{nm}^\pm(E)$:

$$I_\beta = \lim_{\eta \to 0} \left\langle |G_{nm}^+(E) G_{nm}^-(E)|^\beta \right\rangle, \quad 0 < \beta < 1/2. \tag{20}$$

Now, using the functional representation Eq. (5) of retarded and advanced Green's functions, introducing average the product $\left[ G_{nn}^+(E) \right]^\beta \left[ G_{nn}^-(E) \right]^\beta$ over the diagonal disorder and $a = 1, 2, \ldots, \beta$ copies of the system we express:

$$\left\langle \left[ G_{nl}^+(E) \right]^\beta \left[ G_{nl}^-(E) \right]^\beta \right\rangle = \int \left( \prod_{ka} [d\Phi_k^{(a)} d\Phi_k^{(a)\dagger}] \right) \prod_a \chi_R^{(a)*}(l) \chi_R^{(a)}(n) \chi_A^{(a)}(n) \chi_A^{(a)*}(l) \, e^{-S[\Phi, \Phi^\dagger]}, \tag{21}$$

$$S[\Phi, \Phi^\dagger] = -\sum_m \ln \tilde{F} \left( \sum_a \Phi_m^{(a)\dagger} L \Phi_m^{(a)} \right) - i \sum_{mka} \Phi_m^{(a)\dagger} L \left( E\delta_{mk} - T_{mk} + i\eta\Lambda\delta_{mk} \right) \Phi_k^{(a)}. \tag{22}$$

## 3.1 Integration over the phases

Let us first integrate over phases $\phi_{R/A}^{(a)}(k)$ of complex variables $S_{R/A}^{(a)}(k)$. To accomplish that, one can first notice that $T$ is real and symmetric. Thus, one can rearrange the hopping term in the action as

$$
\sum_{mka} T_{mk} \Phi_m^{(a)\dagger} L \Phi_k^{(a)} = \sum_{mka} T_{mk} \operatorname{Re}\{ S_R^{(a)*}(m) S_R^{(a)}(k)
$$
$$
- S_A^{(a)*}(m) S_A^{(a)}(k)\} + \sum_{mka} T_{mk} \left( \chi_R^{(a)*}(m) \chi_R^{(a)}(k) + \chi_A^{(a)*}(m) \chi_A^{(a)}(k) \right), \quad (23)
$$

which suggests to switch to the modulus and the phase of the complex variables $S_{R/A}^{(a)}(k) = |S_{R/A}^{(a)}(k)| e^{i\phi_{R/A}^{(a)}(k)}$. The non-Grassmann part of the hopping term can be written as

$$
\sum_{mka} T_{mk} \operatorname{Re}\{ S_R^{(a)*}(m) S_R^{(a)}(k) - S_A^{(a)*}(m) S_A^{(a)}(k)\}
$$
$$
= \sum_{mka} T_{mk} \left\{ |S_R^{(a)}(m)||S_R^{(a)}(k)| \cos\left( \phi_R^{(a)}(m) - \phi_R^{(a)}(k) \right) - (R \to A) \right\}. \quad (24)
$$

Then evaluating integrals over phases and introducing new variables:

$$
\left| S_R^{(a)}(k) \right|^2 \equiv s_k^{(a)}, \quad \left| S_A^{(a)}(k) \right|^2 \equiv \tilde{s}_k^{(a)}, \quad (25)
$$

we obtain:

$$
\langle \left[ G_{nl}^+(E) \right]^\beta \left[ G_{nl}^-(E) \right]^\beta \rangle = \int \left( \prod_{ka} ds_k^{(a)} d\tilde{s}_k^{(a)} d\chi_R^{(a)}(k) d\chi_R^{(a)*}(k) d\chi_A^{(a)*}(k) d\chi_A^{(a)}(k) \right) e^{-S_1[s,\chi]} \times
$$
$$
\times \prod_a \chi_R^{(a)*}(l) \chi_R^{(a)}(n) \chi_A^{(a)}(n) \chi_A^{(a)*}(l) \times
$$
$$
\times \prod_{\langle km \rangle} J_0\left( 2\sqrt{s_k^{(a)} s_m^{(a)}} \right) J_0\left( 2\sqrt{\tilde{s}_k^{(a)} \tilde{s}_m^{(a)}} \right) e^{-i\left( \chi_R^{(a)*}(m) \chi_R^{(a)}(k) + \chi_A^{(a)*}(m) \chi_A^{(a)}(k) + sym. \right)}, \quad (26)
$$

where the diagonal part of the action reads as

$$
S_1[s,\chi] = -\sum_m \ln \tilde{F}\left( \sum_a \left[ s_m^{(a)} - \tilde{s}_m^{(a)} + \chi_R^{(a)*}(m) \chi_R^{(a)}(m) + \chi_A^{(a)*}(m) \chi_A^{(a)}(m) \right] \right)
$$
$$
- iE \sum_{ma} \left( s_m^{(a)} - \tilde{s}_m^{(a)} + \chi_R^{(a)*}(m) \chi_R^{(a)}(m) + \chi_A^{(a)*}(m) \chi_A^{(a)}(m) \right) + \eta \sum_{ma} \left( s_m^{(a)} + \tilde{s}_m^{(a)} \right). \quad (27)
$$

We have dropped the Grassmann term with $\eta$ in front of it.

## 3.2 Integration over anti-commuting variables

Next, we perform a shift of variables

$$
s_m^{(a)} \to s_m^{(a)} - \chi_R^{(a)*}(m) \chi_R^{(a)}(m),
$$
$$
\tilde{s}_m^{(a)} \to \tilde{s}_m^{(a)} + \chi_A^{(a)*}(m) \chi_A^{(a)}(m). \quad (28)
$$

The domain of integration is correctly captured if one writes down the transformed step functions in the integration measure explicitly as

$$
\theta(s_m^{(a)}) \to \theta(s_m^{(a)}) - \delta(s_m^{(a)}) \chi_R^{(a)*}(m) \chi_R^{(a)}(m),
$$
$$
\theta(\tilde{s}_m^{(a)}) \to \theta(\tilde{s}_m^{(a)}) + \delta(\tilde{s}_m^{(a)}) \chi_A^{(a)*}(m) \chi_A^{(a)}(m). \quad (29)
$$

The advantage of this shift is that it removes the anti-commuting variables from the diagonal part of the action in (27). The integration over anti-commuting variables in Eqs. (26), (27) factorizes as $M = \prod_a M_a^R M_a^A$, where

$$M_a^R = \int \prod_k \left( d\chi_R^{(a)}(k) d\chi_R^{(a)*}(k) \right) \chi_R^{(a)*}(l) \chi_R^{(a)}(n) \prod_m \left( \theta(s_m^{(a)}) - \delta(s_m^{(a)}) \chi_R^{(a)*}(m) \chi_R^{(a)}(m) \right) \times$$

$$\times \prod_{\langle km \rangle} J_0 \left( 2\sqrt{(s_k^{(a)} - \chi_R^{(a)*}(k)\chi_R^{(a)}(k))(s_m^{(a)} - \chi_R^{(a)*}(m)\chi_R^{(a)}(m))} \right) e^{-i\left( \chi_R^{(a)*}(m)\chi_R^{(a)}(k) + sym. \right)}, \quad (30)$$

and

$$M_a^A = \int \prod_k \left( d\chi_A^{(a)*}(k) d\chi_A^{(a)}(k) \right) \chi_A^{(a)}(n) \chi_A^{(a)*}(l) \prod_m \left( \theta(\tilde{s}_m^{(a)}) + \delta(\tilde{s}_m^{(a)}) \chi_A^{(a)*}(m) \chi_A^{(a)}(m) \right) \times$$

$$\times \prod_{\langle km \rangle} J_0 \left( 2\sqrt{(\tilde{s}_k^{(a)} + \chi_A^{(a)*}(k)\chi_A^{(a)}(k))(\tilde{s}_m^{(a)} + \chi_A^{(a)*}(m)\chi_A^{(a)}(m))} \right) e^{-i\left( \chi_A^{(a)*}(m)\chi_A^{(a)}(k) + sym. \right)}. \quad (31)$$

Let us assume that the initial point $n = 1$ is a root of a tree. All the anti-commuting variables in the integral Eq. (30) are divided in two parts: those which correspond to sites on the (unique) path $\mathcal{P}_l$ from the root to the final point $l$ and the remaining variables. Let us first consider integration over the anti-commuting variables of the first part. The result is given only by the saturated (i.e. that have exactly one pair of variables $\chi \chi^*$ corresponding to any copy $(a)$ and any site $p$) set of variables in the expansion of exponent in Eq. (30) and the "source" variables $\chi_{R/A}^{(a)*}(l)\chi_{R/A}^{(a)}(n)$. The integration over retarded variables gives $i^{|\mathcal{P}_l|}$, and the advanced components produce the factor $(-i)^{|\mathcal{P}_l|}$ (where $|\mathcal{P}_l|$ is the length of the path), so they compensate each other (note the order of anti-commuting variables in the measure and in the pre-factor!).

Notice that, since the integrals along the path $\mathcal{P}_l$ are already saturated, the hopping terms in the exponent of Eqs. (30), (31) connecting the remaining branches of the tree and the path $\mathcal{P}_l$ can be omitted. Therefore, our final step is to integrate over all possible anti-commuting variables of "decoupled" branches. This task can be accomplished iteratively.

Let us start from the boundary. We are going to use the following short notation: $\chi^*$ and $\chi$ will denote the anti-commuting variables that we are currently integrating over, while $\zeta^*$ and $\zeta$ will stand for the variables belonging to the unique predecessor of the chosen site. Therefore, we need to compute the following integral:

$$\tilde{\Xi}_0(s,s') = \int d\chi \, d\chi^* \left[ \theta(s') + \delta(s')\chi^*\chi \right] \quad (32)$$

$$\times J_0 \left( 2\sqrt{ss'} + \left( \sqrt{\frac{s'}{s}}\zeta^*\zeta + \sqrt{\frac{s}{s'}}\chi^*\chi \right) + \frac{1}{2\sqrt{ss'}}\chi^*\chi\zeta^*\zeta \right) e^{-i\zeta^*\chi - i\chi^*\zeta}, \quad (33)$$

where $e^{-i\zeta^*\chi - i\chi^*\zeta} = 1 - i(\zeta^*\chi + \chi^*\zeta) + \chi^*\chi\zeta^*\zeta$. The integral Eq. (32) is equal, up to an overall opposite sign, to a single integral in $M_a^A$. A remarkable fact is that this integral, in fact, does not depend on $\zeta$ and $\zeta^*$.

Indeed, expanding the Bessel functions in the anti-commuting variables, combining the result with the expansion of the exponential term and using the identity for Bessel functions $J_2(x) - J_0(x) - 2\frac{J_1(x)}{x} = -2J_0(x)$ one can easily see that the contribution proportional to $\zeta^*\zeta$ is canceled out, and the remaining integral leads to

$$\tilde{\Xi}_0(s,s') = \delta(s') - \theta(s')\sqrt{\frac{s}{s'}} J_1 \left( 2\sqrt{ss'} \right). \quad (34)$$

The integral over retarded variables can be performed in a similar way. As a result, the total contribution from the anti-commuting variables has the following form:

$$M = \prod_a \prod_{\substack{\langle mk \rangle \\ m,k \in \mathcal{P}_l}} \left( J_0\left(2\sqrt{s_m^{(a)}s_k^{(a)}}\right) J_0\left(2\sqrt{\tilde{s}_m^{(a)}\tilde{s}_k^{(a)}}\right) \right) \prod_{\substack{\langle mk \rangle \\ m,k \notin \mathcal{P}_l}} \left( \tilde{\Xi}_0(s_m^{(a)}, s_k^{(a)}) \tilde{\Xi}_0(\tilde{s}_m^{(a)}, \tilde{s}_k^{(a)}) \right). \quad (35)$$

Note that $\tilde{\Xi}_0(s_k, s_m)$ is not symmetric, so we always assume that the site $k$ (corresponding to the first argument $s_k$) is closer to the root than the site $m$ corresponding to the second argument $s_m$.

Finally, one can represent the average of interest solely in terms of the conventional integrals over $s_k^{(a)}$ and $\tilde{s}_k^{(a)}$ as:

$$\langle \left[ G_{nl}^+(E) \right]^\beta \left[ G_{nl}^-(E) \right]^\beta \rangle = \int \left( \prod_{ka} ds_k^{(a)} d\tilde{s}_k^{(a)} \theta(s_k^{(a)}) \theta(\tilde{s}_k^{(a)}) \right) e^{-S_1[s,0]} \times$$

$$\times \prod_a \prod_{\substack{\langle mk \rangle \\ m,k \in \mathcal{P}_l}} \left( J_0\left(2\sqrt{s_m^{(a)}s_k^{(a)}}\right) J_0\left(2\sqrt{\tilde{s}_m^{(a)}\tilde{s}_k^{(a)}}\right) \right) \prod_{\substack{\langle mk \rangle \\ m,k \notin \mathcal{P}_l}} \left( \tilde{\Xi}_0(s_m^{(a)}, s_k^{(a)}) \tilde{\Xi}_0(\tilde{s}_m^{(a)}, \tilde{s}_k^{(a)}) \right), \quad (36)$$

where $a = 1, 2, \ldots, \beta$. At this moment, we are left with $2\beta$ real variables at each node. The goal of the next section is to reduce this set to a single variable, $\beta$ appearing as a parameter in the integrand. If this goal is accomplished, one can easily make an analytic continuation over $\beta$.

## 3.3 Integration over $s_k$ and $\tilde{s}_k$

Our next step is to integrate over $s_k$ and $\tilde{s}_k$. This can be done by means of the following identity

$$1 = \prod_m \int_{-\infty}^{+\infty} dv_m \, \delta\left(v_m - \sum_a \left[ s_m^{(a)} - \tilde{s}_m^{(a)} \right]\right) = \prod_m \int_{-\infty}^{+\infty} \frac{dv_m dz_m}{2\pi} \, e^{iz_m\left(v_m - \sum_a \left[ s_m^{(a)} - \tilde{s}_m^{(a)} \right]\right)}. \quad (37)$$

In terms of these new variables, the action reads as

$$S_1[s,0] = -\sum_m \ln \tilde{F}(v_m) - iE \sum_m v_m + \eta \sum_{ma} \left( s_m^{(a)} + \tilde{s}_m^{(a)} \right). \quad (38)$$

Now we are in a position to integrate over $s_k$ and $\tilde{s}_k$ because they are decoupled from each other. The integration can again be performed iteratively, starting from the boundary (once again, we assume that $n \equiv 1$ is the root of the tree).

First of all, one can easily verify that:

$$\int_0^{+\infty} ds_k^{(a)} \, e^{-i(z_k - i\eta)s_k^{(a)}} \tilde{\Xi}_0\left(s_m^{(a)}, s_k^{(a)}\right) = e^{\frac{is_m^{(a)}}{z_k - i\eta}}. \quad (39)$$

Thus, the complete integration over $2\beta$ initial real variables for a given node at the boundary is given by

$$e^{\frac{iz_k}{z_k^2 + \eta^2} \sum_a \left[ s_m^{(a)} - \tilde{s}_m^{(a)} \right] - \frac{\eta}{z_k^2 + \eta^2} \sum_a \left[ s_m^{(a)} + \tilde{s}_m^{(a)} \right]}, \quad (40)$$

where $\sum_a \left[ s_m^{(a)} - \tilde{s}_m^{(a)} \right]$ can be replaced by $v_m$ due to the presence of the delta function (37). Moreover, as it will be explained later, one can safely set $\eta = 0$.

Next, by combining this new term with $e^{iz_k v_k}$ and integrating over $z_k$, we obtain the effective kernel operating on a given link which does not belong to the path $\mathcal{P}_l$:

$$
\Xi_0(v_m, v_k) = \frac{1}{2\pi} \int\limits_{-\infty}^{+\infty} dz \, e^{i(vz^{-1} + v'z)} = \frac{1}{\pi} \int\limits_{0}^{+\infty} dz \, \cos\left(vz^{-1} + v'z\right)
$$

$$
= \delta(v_k) - \theta(v_k v_m) \sqrt{\frac{|v_m|}{|v_k|}} J_1\left(2\sqrt{|v_m||v_k|}\right). \tag{41}
$$

Remarkably, it coincides with the kernel of the non-linear equation (16). This process can be continued further, involving other links that do not belong to the path $\mathcal{P}_l$. The integration along the path $\mathcal{P}_l$ is only slightly different. We make use of the following integral

$$
\int\limits_{0}^{+\infty} ds_k^{(a)} \, e^{-i(z_k - i\eta)s_k^{(a)}} J_0\left(2\sqrt{s_m^{(a)} s_k^{(a)}}\right) = \frac{-i}{z_k - i\eta} e^{\frac{is_m^{(a)}}{z_k - i\eta}}. \tag{42}
$$

Therefore, the integration over $s_k^{(a)}$ leads to

$$
\frac{1}{(z_k^2 + \eta^2)^\beta} e^{\frac{iz_k}{z_k^2 + \eta^2} \sum_a \left[s_m^{(a)} - \tilde{s}_m^{(a)}\right] - \frac{\eta}{z_k^2 + \eta^2} \sum_a \left[s_m^{(a)} + \tilde{s}_m^{(a)}\right]}, \tag{43}
$$

where $\sum_a \left[s_m^{(a)} - \tilde{s}_m^{(a)}\right]$ can again be replaced by $v_m$ due to the presence of the delta function (37). We also denote here $l_m = \sum_a \left[s_m^{(a)} + \tilde{s}_m^{(a)}\right]$.

By combining the resulting expression with $e^{iz_k v_k}$, we obtain the following integral over $z_k$

$$
\mathcal{R}(v_m, l_m | v_k) = \int\limits_{-\infty}^{+\infty} \frac{dz_k}{(z_k^2 + \eta^2)^\beta} e^{\frac{iz_k v_m}{z_k^2 + \eta^2} - \frac{\eta l_m}{z_k^2 + \eta^2} + iz_k v_k}. \tag{44}
$$

Crucially, this expression is an analytic function of $\beta$, and thus, we are in a position to analytically continue it to the region $\beta < 1/2$. This procedure makes the integral over $z_k$ convergent even at $\eta = 0$ and justifies the limit $\eta = 0$ in Eqs. (40), (43) which eliminates the dependence on $l_m$ whatsoever. Indeed, in the course of the subsequent integration over $s_m$ and $\tilde{s}_m$ the dominant region $s_m, \tilde{s}_m \sim z_m^{-1}$ is finite in the limit $\eta \to 0$ due to the convergence of the integral over $z_m$.

One obtains the effective $\beta$-dependent kernel operating on a given link belonging to the path $\mathcal{P}_l$:

$$
\Xi_\beta(v, v') = \frac{1}{\pi} \int\limits_{0}^{+\infty} \frac{dz}{|z|^{2\beta}} \cos\left(vz^{-1} + v'z\right), \tag{45}
$$

which coincides with the kernel (13) in the linear eigenvalue problem Eq. (14).

This integral can be evaluated exactly as follows

$$
\Xi_\beta(v, v') = \frac{2}{\pi} \left(\frac{|v|}{|v'|}\right)^{\frac{1}{2} - \beta} \left\{ \theta(-vv') \sin(\pi\beta) K_{1-2\beta}\left(2\sqrt{|v'||v|}\right) + \right.
$$

$$
\left. + \frac{\pi \theta(vv')}{4\cos(\pi\beta)} \left(J_{2\beta-1}\left(2\sqrt{|v'||v|}\right) - J_{1-2\beta}\left(2\sqrt{|v'||v|}\right)\right) \right\}, \tag{46}
$$

where $K_m(x)$ is the modified Bessel function of the second kind of $m$th order.

The last remaining integration over the root $n = 1$ can be performed in the same way as in Eq. (42) by setting all $s_m^{(a)}$ to zero (since the root has no predecessor). Thus, the integral over $z_1$ leads to

$$\frac{1}{2\pi} \int_{-\infty}^{+\infty} \frac{dz_1}{(z_1^2 + \eta^2)^\beta} e^{iz_1 v_1} = \frac{|v_1|^{2\beta-1}}{2\cos(\pi\beta)\Gamma(2\beta)}, \quad \eta \to 0^+. \tag{47}$$

The final expression reads as follows:

$$\langle \left[G_{nl}^+(E)\right]^\beta \left[G_{nl}^-(E)\right]^\beta \rangle = \frac{1}{2\cos(\pi\beta)\Gamma(2\beta)} \int_{-\infty}^{+\infty} \left(\prod_k dv_k\, \tilde{F}(v_k)e^{iEv_k}\right)|v_1|^{2\beta-1} \times$$

$$\times \prod_{\substack{\langle mk \rangle \\ m,k \in \mathcal{P}_l}} \Xi_\beta(v_m, v_k) \prod_{\substack{\langle mk \rangle \\ m,k \notin \mathcal{P}_l}} \Xi_0(v_m, v_k). \tag{48}$$

## 3.4 Iterative representation of the result

The multiple integral Eq. (48) can be represented in a form of iterations, similar to the TM equation. To this end we introduce two functions $\Psi_0^{(r)}(v)$ and $\Psi_\beta^{(r)}(v)$ obeying the following recursive equations:

$$\Psi_0^{(r+1)}(v) = \int_{-\infty}^{+\infty} dv'\, \Xi_0(v, v')\tilde{F}(v')e^{iEv'}[\Psi_0^{(r)}(v')]^K, \tag{49}$$

with the initial condition $\Psi_0^{(0)}(v) \equiv 1$, and

$$\Psi_\beta^{(r+1)}(v) = \int_{-\infty}^{+\infty} dv'\, \Xi_\beta(v, v')\tilde{F}(v')e^{iEv'}\Psi_\beta^{(r)}(v') \left[\Psi_0^{(R-|\mathcal{P}_l|-1+r)}(v')\right]^{K-1}, \tag{50}$$

with the initial condition $\Psi_\beta^{(0)}(v) \equiv \Psi_0^{(R-|\mathcal{P}_l|-1)}(v)$. Here $R$ is the total number of generations on the tree and $|\mathcal{P}_l|$ is the length of the path $\mathcal{P}_l$.

The function $\Psi_0^{(r)}(v)$ describes the summation over the tree branch with $r$ generations which the path $\mathcal{P}_l$ does not belong to. One can immediately recognize in Eq. (49) the non-linear equation, Eq. (12), for the zero-order approximation of the TM equation with the same initial condition. Thus the self-consistent solution to Eq. (49) is $\Psi_0(v) \equiv \Omega_0(v)$. On the other hand, the function $\Psi_\beta^{(r)}(v)$ is related to the summation over the part of the path $\mathcal{P}_l$ of the length $r$.

Then the result of the previous subsection Eq. (48) can be expressed through these functions as:

$$\langle |G_{1l}(E)|^{2\beta} \rangle = \frac{1}{2\cos(\pi\beta)\Gamma(2\beta)} \int_{-\infty}^{+\infty} dv\, |v|^{2\beta-1}\tilde{F}(v)e^{iEv}\Psi_\beta^{(|\mathcal{P}_l|)}(v) \left[\Psi_0^{(R-1)}(v)\right]^{K-1}. \tag{51}$$

Let us now assume the separation of scales with the following order $1 \ll |\mathcal{P}_l| \ll R$. Then, after many consequent integrations, the generating function $\Psi_0^{(R-|\mathcal{P}_l|-1)}(v)$ can be replaced by

$\Omega_0(v)$ which is the self-consistent solution of Eq. (12). Moreover, in the integrations along $\mathcal{P}_l$, only the right eigenfunction with the largest eigenvalue $\epsilon_\beta = \max_\kappa \{\epsilon_\beta^{(\kappa)}\}$ survives, which can be found as usual from the equation:

$$\epsilon_\beta^{(\kappa)} f_\beta^{(\kappa)}(v) = \int\limits_{-\infty}^{+\infty} dv' \, \Xi_\beta(v, v') \tilde{\mathcal{F}}(v') e^{iEv'} f_\beta^{(\kappa)}(v'), \tag{52}$$

where $\tilde{\mathcal{F}}$ is the renormalized Fourier-transform of the on-site disorder distribution obeying Eq. (15). Therefore

$$\Psi_\beta^{(|\mathcal{P}_l|)}(v) \approx (\epsilon_\beta)^{|\mathcal{P}_l|} C_\beta f_\beta(v), \tag{53}$$

where $C_\beta$ is the coefficient in the decomposition of the initial condition $\Omega_0(v)$ in terms of the left eigenfunctions

$$C_\beta = \int\limits_{-\infty}^{+\infty} dv \, g_\beta(v) \Omega_0(v), \tag{54}$$

and $g_\beta(v)$ corresponds to the same largest eigenvalue

$$\epsilon_\beta g_\beta(v) = \tilde{\mathcal{F}}(v) e^{iEv} \int\limits_{-\infty}^{+\infty} dv' \, g_\beta(v') \Xi_\beta(v', v), \tag{55}$$

and we assumed that the there is a finite gap between the largest eigenvalue $\epsilon_\beta$ and the second largest eigenvalue. Finally, we obtain

$$\langle |G_{1l}(E)|^{2\beta} \rangle = (\epsilon_\beta)^{|\mathcal{P}_l|} \chi_\beta, \quad |\mathcal{P}_l|, R \to \infty, \quad |\mathcal{P}_l|/R \to 0, \tag{56}$$

where

$$\chi_\beta = \frac{C_\beta}{2\cos(\pi\beta)\Gamma(2\beta)} \int\limits_{-\infty}^{+\infty} dv \, |v|^{2\beta-1} \tilde{\mathcal{F}}(v) e^{iEv} f_\beta(v). \tag{57}$$

Eqs. (56), (57) is the main result of Section III and the main technical result of this paper.

## 4 Physical meaning of $\Omega_0(v)$.

In this section we establish the physical meaning of the renormalization factor $\Omega_0(v)$ in Eq. (15). To this end we use Eq. (51) to compute the distribution function of $G_{1,1}$ in the root of a tree. In this particular case $\Psi_\beta^{(|\mathcal{P}_l|)}(v)$ and $\left[\Psi_0^{(R-1)}(v)\right]^{K-1}$ combine together to give $[\Omega_0(v)]^K$.

Next we compute the distribution function of a real $g \equiv G_{1,1}$ by a Mellin transform:

$$P(g) = \frac{1}{g} \int_B \frac{d\beta}{\pi i} \exp[-2\beta \ln g] M_\beta, \tag{58}$$

where $M_\beta$ is the moment of $G_{1,1}^2$ found from Eq. (51) and $B$ is the standard Bromwich contour $[c - i\infty, c + i\infty]$, with a real $0 < c < 1/2$. Thus on the initial Bromwich contour $\operatorname{Re}\beta < 1/2$, and Eq. (51) holds true. The integral over $\beta$ that emerges in $P(g)$ is the following:

$$\varphi(z) = \int_B \frac{d\beta}{2\pi i} \frac{\exp[2\beta z]}{\cos(\pi\beta)\Gamma(2\beta)}, \quad e^z = |v|/g. \tag{59}$$

Note that the $\Gamma(2\beta)$ function in the denominator at large $|\beta| \gg 1$ is larger than any exponent at $\mathrm{Re}\,\beta > 0$ and it is smaller than any exponent at $\mathrm{Re}\,\beta < 0$. Therefore, *at any $z$* the Bromwich contour can be deformed so to surround the poles of the analytically continued integrand, located at $\beta = 1/2 + k$, $k = 0, 1, 2, \dots$. Then the integral Eq. (59) is given by the sum of residues in these poles:

$$\varphi(z) = \sum_{k=0}^{\infty} \frac{e^{(1+2k)z}(-1)^k}{(2k)!} = e^z \cos(e^z) = (|v|/g)\cos(v/g), \tag{60}$$

and the distribution function $P(g)$ is given by:

$$P(x = 1/g) = g^2 P(x = g) = \int_{-\infty}^{+\infty} dv\, e^{iv/g}\, \tilde{F}(v)\,[\Omega_0(v)]^K. \tag{61}$$

Here for simplicity we consider $E = 0$ and a symmetric distribution of on-site energies with the symmetric Fourier transform $\tilde{F}(v) = \tilde{F}(-v)$. In this case in Eq. (61) one can replace $\cos(|v|/g) \to e^{iv/g}$.

Eq. (61) can be considered as the distribution of the sum of $K$ i.i.d. quantities $g_i$, $i = 1, 2, \dots K$ with the identical distribution functions $f(g_i)$ given by the Fourier transform of $\Omega_0(v)$. Indeed, let us multiply the identity:

$$\int \delta\left(\zeta - \sum_{i=1}^{K} g_i\right) \prod_{i=1}^{K} f(g_i)\, dg_i = \int \frac{dv}{2\pi} e^{i\zeta v}\left(\int e^{-ivx} f(x)\, dx\right)^K, \tag{62}$$

by $F(\zeta - g^{-1})$, set $\zeta = \varepsilon + g^{-1}$ and integrate over the on-site energy $\varepsilon$. Then the on-site energy distribution $F(\varepsilon)$ will be Fourier-transformed $F(\varepsilon) \to \tilde{F}(v)$ and the r.h.s. of Eq. (62) takes the form of the r.h.s. of Eq. (61) with $\Omega_0(v)$ being the Fourier transform of $f(x)$. Equating the l.h.s. of Eqs. (61), (62) we obtain:

$$P(x = 1/g) = \int \delta\left(g^{-1} + \varepsilon - \sum_{i=1}^{K} g_i\right) \prod_{i=1}^{K} f(g_i)\, dg_i. \tag{63}$$

The argument of the $\delta$-function gives us immediately the physical meaning of the quantities $g_i$ as the one-point (cavity) Green's functions at the $K$ sites which the site 1 is the descendant of, like in the Abou-Chacra-Thouless-Anderson equation, $g^{-1} = -\varepsilon + \sum_{i=1}^{K} g_i$ (cf. Eq. (82)). Correspondingly, the function $f(g_i)$ is the distribution function of such Green's functions. This gives the physical meaning of $\Omega_0(v)$

$$\Omega_0(v) = \int e^{-iv g_i} f(g_i)\, dg_i, \tag{64}$$

as the Fourier-transform of the distribution functions of a real one-point cavity Green's function $g_i$.

This result explains why in the non-linear sigma-model on a Cayley tree $\Omega_0(v) = 1$, while on the tree with one orbital per site it is a non-trivial function of $v$. One should remember that the nonlinear sigma-model in the problem of localization was derived by Wegner [34] in the limit of an infinite number of states (orbitals) per site. Physically, this corresponds to a model in which a site is represented by a granule with a very large number of states in it. Thus the "one-point Green's function" in this model is given by a sum of one-point Green's functions in a granule divided by the number of states in it. In the limit of an infinite number of states in a granule, the distribution of such a quantity is a delta-function, and its Fourier-transform, $\Omega_0(v) \equiv 1$. At any finite number of orbitals per site the delta-function is broadened and for just one orbital per site $f(x)$ and its Fourier-transform $\Omega_0(v)$ are non-trivial functions of $x$ and $v$.

## 5 Strong-disorder approximations for $\epsilon_\beta$.

In this section we derive simple approximations for $\epsilon_\beta$ and control their accuracy. The idea is that at strong disorder the renormalization of the on-site energy distribution is weak and at sufficiently large $W$ can be neglected. In order to show this we note that the integral part of Eq. (12) converges at $|v'| \sim W^{-1} \ll 1$ due to the factor $\tilde{F}(v')$. The argument of function $\Omega_0(v)$ entering the spectral problem Eq. (14) is also effectively restricted by a similar factor in this equation. This allows to expand the Bessel function in the kernel of Eq. (12) and represent the kernel as a sum of the factorized ones which makes the equation solvable. Expansion of the Bessel function in Eq. (12) to the lowest order in its argument then after linearization leads to the solution:

$$\Omega_0(v) \approx 1 - |v|\frac{\int dv' \,\tilde{F}(v')}{1 + K \int dv' |v'| \tilde{F}(v')} = 1 + O(1/W^2). \tag{65}$$

This solution shows that the renormalized distribution $\mathcal{F}(\varepsilon)$, Eq. (15), acquires a $1/\varepsilon^2$ tail for any bare distribution (with the convergent mean value) which decreases at large $\varepsilon$ faster than or similar to $1/\varepsilon^2$. In this paper we consider three such distributions, the box-shaped distribution, the Gauss distribution and the Cauchy distribution which Fourier transforms read as follows:

$$F_{\rm b}(\varepsilon) = \frac{\theta(|\varepsilon| - W/2)}{W}, \ \ \tilde{F}_{\rm b}(v) = \frac{2\sin(W v/2)}{(W v)}, \tag{66}$$

$$F_{\rm G}(\varepsilon) = \frac{1}{\sqrt{\pi W^2}} e^{-\varepsilon^2/W^2}, \ \ \tilde{F}_{\rm G}(v) = e^{-(W/4)^2 v^2}, \tag{67}$$

$$F_{\rm C}(\varepsilon) = \frac{(W/2\pi)}{\varepsilon^2 + (W/2)^2}, \ \ \tilde{F}_{\rm C}(v) = e^{-(W/2)|v|}. \tag{68}$$

As is said before, any of these bare distributions acquire a tail $A/\varepsilon^2$; however, the prefactor $A$ is small, e.g. for the box distribution:

$$A_{\rm box} = \frac{\pi}{W + 4K/W}. \tag{69}$$

The simplest approximation at large disorder is to neglect this renormalization whatsoever: $\Omega_0 = 1$.

With this assumption, expanding the Bessel functions in the kernel of Eq. (52), one obtains a $2 \times 2$ matrix eigenvalue problem:

$$\epsilon_\beta I_1 = \frac{\sin(\pi\beta)}{\pi}\Gamma(1-2\beta)I_2 \int dv' \,\tilde{F}(v') + \frac{\sin(\pi\beta)}{\pi}\Gamma(2\beta-1)I_1 \int dv' \,\tilde{F}(v')\,|v'|^{1-2\beta},$$

$$\epsilon_\beta I_2 = \frac{\sin(\pi\beta)}{\pi}\Gamma(1-2\beta)I_2 \int dv' \,\tilde{F}(v')|v'|^{2\beta-1} + \frac{\sin(\pi\beta)}{\pi}\Gamma(2\beta-1)I_1 \int dv' \,\tilde{F}(v'),$$

where

$$I_1 = \int dv \,\tilde{F}(v) f_\beta(v), \quad I_2 = \int dv \,|v|^{2\beta-1}\tilde{F}(v) f_\beta(v).$$

One can see that in this approximation there are two eigenvalues of which the largest is:

- For the box distribution:

$$\epsilon_\beta = \frac{1}{W\,|1-2\beta|} \left\{ \text{sign}(1-2\beta)\left[\left(\frac{W}{2}\right)^{1-2\beta} - \left(\frac{W}{2}\right)^{2\beta-1}\right] \right.$$
$$\left. + \sqrt{\left[\left(\frac{W}{2}\right)^{1-2\beta} + \left(\frac{W}{2}\right)^{2\beta-1}\right]^2 - 2\pi(1-2\beta)\tan(\pi\beta)} \right\}. \tag{70}$$

- For the Gaussian distribution:

$$\epsilon_\beta = -\frac{\sqrt{\pi}}{\left(\frac{1}{2}-\beta\right)\cos(\pi\beta)W} \left\{ \frac{\left(\frac{W}{2}\right)^{1-2\beta}}{\Gamma\left(\beta-\frac{1}{2}\right)} + \frac{\left(\frac{W}{2}\right)^{2\beta-1}}{\Gamma\left(\frac{1}{2}-\beta\right)} \right.$$
$$\left. - \sqrt{\left[\frac{\left(\frac{W}{2}\right)^{1-2\beta}}{\Gamma\left(\beta-\frac{1}{2}\right)} - \frac{\left(\frac{W}{2}\right)^{2\beta-1}}{\Gamma\left(\frac{1}{2}-\beta\right)}\right]^2 - \frac{1-2\beta}{\pi}\sin(2\pi\beta)} \right\}. \tag{71}$$

- For the Cauchy distribution:

$$\epsilon_\beta = \frac{1}{W\,|\cos(\pi\beta)|} \left\{ \text{sign}(1-2\beta)\left[\left(\frac{W}{2}\right)^{1-2\beta} - \left(\frac{W}{2}\right)^{2\beta-1}\right] \right.$$
$$\left. + \sqrt{\left[\left(\frac{W}{2}\right)^{1-2\beta} + \left(\frac{W}{2}\right)^{2\beta-1}\right]^2 - \frac{4\sin(2\pi\beta)}{\pi(1-2\beta)}} \right\}. \tag{72}$$

One can see that all those expressions for $\epsilon_\beta$ respect the basic symmetry discovered in a seminal work of Abou-Chacra, Thouless and Anderson (ACTA) [24]:

$$\epsilon_\beta = \epsilon_{1-\beta}. \tag{73}$$

Furthermore, Eqs. (70)-(72) respect the exact property of $\epsilon_\beta$:

$$\epsilon_{\beta=0} = 1. \tag{74}$$

Note that earlier in Refs. [25, 28] there was proposed another approximation for $\epsilon_\beta$ at large disorder for the box distribution:

$$\epsilon_\beta \approx \frac{(W/2)^{1-2\beta} - (W/2)^{2\beta-1}}{(W/2-2/W)(1-2\beta)}. \tag{75}$$

The accuracy of this simple approximation is $O(W\ln W)^{-1}$, while the accuracy of Eqs. (70) and (71) is much higher (see Fig. 3 and Fig. 4):

$$\Delta\epsilon_{\beta=1/2} = O(W^2\ln W)^{-1}. \tag{76}$$

Surprisingly, for the Cauchy distribution Eq. (72) gives incredibly high accuracy for the localization transition point which is known exactly from Ref. [24]:

$$(4/\pi)K\,\ln[W_c/2] + 4K^2/(3W_c) = W_c/2. \tag{77}$$

Numerical solution for this equation for $K = 2$ and $K = 3$ gives $W_c/2 = 4.36223$ and $W_c/2 = 9.09131$, respectively. The localization transition point can be also found from the equation [24]:

$$\epsilon_{\beta=1/2}(W) = 1/K. \tag{78}$$

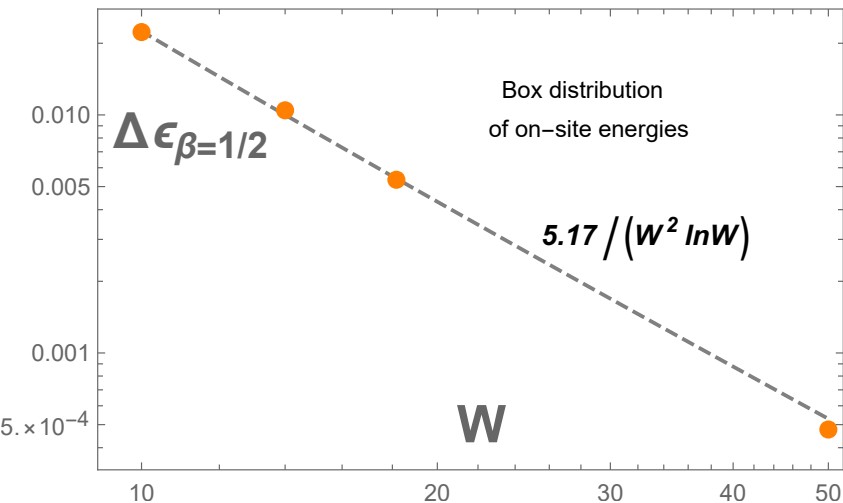

Figure 3: **The accuracy of approximation for $\epsilon_\beta$, Eq. (70), for the box distribution of on-site energies at large disorder $W \gg 1$.** The difference $\Delta\epsilon_\beta$ between the numerical solution for $\epsilon_{\beta=1/2}$ of the exact Eqs. (12), (14) and the approximate solution for $\epsilon_{\beta=1/2}$, Eq. (70), as a function of disorder strength $W$ for a $K = 2$ Cayley tree. The dashed gray line is a fit to the function $\Delta\epsilon_{\beta=1/2} = 5.17/(W^2 \ln W)$.

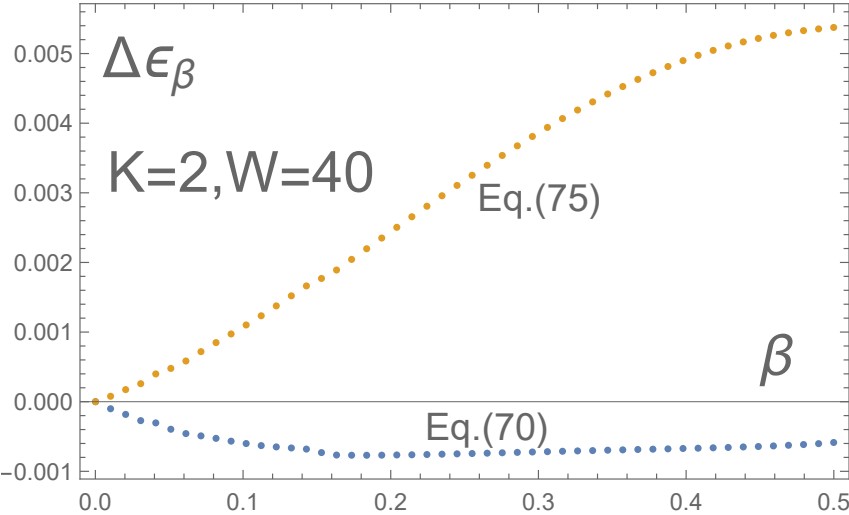

Figure 4: **Comparison of the accuracy of approximation for $\epsilon_\beta$ by Eq. (70) and by Eq. (75).** $\Delta\epsilon_\beta$ is the difference between the numerical solution for $\epsilon_\beta$ of the exact Eqs. (12), (14) and the approximate solutions for $\epsilon_\beta$, Eq. (70) and (75), as a function of $\beta$ at $W = 40$ for a $K = 2$ Cayley tree.

Solving Eq. (78) with $\epsilon_\beta$ from Eq. (72) we found the following numbers: $W_c/2 = 4.36225$ for $K = 2$ and $W_c/2 = 9.09129$ for $K = 3$. This extra-ordinary accuracy seems to indicate on the accuracy of the approximation, Eq. (72), as high as $(W^4 \ln W)^{-1}$.

For the box probability distribution Eq. (78) with the approximate $\epsilon_\beta$ from Eq. (70) gives $W_c = 18.51$ which is close to the value $W_c = 18.17$ found numerically from the exact $\epsilon_\beta$ ( see also [29, 35]). However, it is not so spectacularly close to the exact value as for the Cauchy distribution.

# 6 Statistics of Green's functions $G_{1,r}$ at large $r$

Given the moments, Eq. (3), one may find at large $r$ the distribution function $P(y = \ln |G|)$ by evaluating the Mellin transform Eq. (58) in the saddle-point approximation [27]. The distribution appears to have a very special form of the *large deviation ansatz*:

$$P(y = \ln |G|) \sim e^{r \, \mathcal{R}\left(-\frac{2y}{r}\right)}, \tag{79}$$

where the function $\mathcal{R}(\mathfrak{g})$ is given by the Legendre transform of $\ln \epsilon_\beta$:

$$\mathcal{R}(\mathfrak{g}) = \ln \epsilon_{\beta(\mathfrak{g})} + \mathfrak{g}\,\beta(\mathfrak{g}), \tag{80}$$

$$\mathfrak{g} = -\partial_\beta [\ln \epsilon_\beta]|_{\beta = \beta(\mathfrak{g})}. \tag{81}$$

Eqs. (79), (80), (81) are valid at large $r$ at any disorder which is encoded into the form of $\beta$-dependence of $\epsilon_\beta$. At small disorder $W/2 \lesssim 1$ important is the $K$-dependent renormalization of on-site energy distribution $\mathcal{F}(\varepsilon)$ in Eq. (52) which results in the $K$-dependence of $\epsilon_\beta$ and $\mathcal{R}(\mathfrak{g})$. At strong disorder $W/2 \gg \sqrt{K-1}$ one can neglect the renormalization of $\mathcal{F}(\varepsilon)$, therefore $\epsilon_\beta$ and $\mathcal{R}(\mathfrak{g})$ are independent of the branching number $K$. In this limit Eqs. (70)-(72) and Eqs. (79)-(81) are equally valid both for the Cayley tree and for the strictly one-dimensional case $K = 1$. In both cases the deviation from FSA result, Eq. (2), is due to resonances along a single shortest path which role is *overestimated* in FSA.

In Fig. 5 we compare: (i) $P(y = \ln |G|)$ for the large-$W$ approximation Eq. (70) (which is indistinguishable from the exact result at $W = 50$); (ii) for the Poisson distribution, Eq. (2), resulting from FSA at the box distribution of on-site energies; and (iii) for the log-normal (Gaussian in $\ln |G|$) distribution which emerges from the Poisson distribution in the large $r$ limit. We consider modestly strong disorder $W = 50$ and $r = 10, 100$.

Fig. 5 demonstrates the main physical result of this paper: FSA fails to describe large deviations $|G| \gg G_{typ}$ from the typical value $G_{typ}$, whereas the region $|G| \lesssim G_{typ}$ it describes quite well.

As a matter of fact, FSA *overestimates* the role of resonances which enhance $|G_{1,r}|$ at large distances $r$. The reason is that FSA, Eq. (1), involves the *on-site energies* $\varepsilon_p$ along the shortest path rather than the exact eigenvalues $E_p$ which emerge due to interference of the "detour"/return paths of the length larger than $r$. Since those paths pass more than one time through the same sites, their amplitudes and the true eigenvalues are *not statistically independent*. To take account of this important point, one should have considered in Eq. (1) the product of exact one-point Green's functions which incorporates the real parts of the self-energies $\Sigma_p$. The large-$W$ approximations developed in this paper fix this drawback in an efficient way. It takes into account the fact that if at some point $p$ of the path the one-point Green's function $\mathcal{G}_p \equiv G_{p,p}$ is large due to resonance $E_p = \varepsilon_p + \Sigma_p \approx E$, it results in a large self energy $\Sigma_{p+1}$ for the Green's function on the next point $p + 1$ of the path according to the Abou-Chacra-Thouless-Anderson equation [24]:

$$\mathcal{G}_{p+1}^{-1} = E - \varepsilon_{p+1} - \Sigma_{p+1}, \quad \Sigma_{p+1} = \sum_{i(p)=1}^{K} \mathcal{G}_{i(p)}, \tag{82}$$

where $i(p)$ denotes $K$ predecessors of $p$. Thus the next Green's function $\mathcal{G}_{p+1}$ must be small which compensates the large value of the preceding Green's function in the product over the path. This mechanism of correlation [28] (which was emphasized by Anderson already in his seminal paper [1]) effectively diminishes the role of resonances and leads to a smaller probability of having the large value of $|G_{1,r}|$, as the Fig. 5 shows. This effect is more pronounced for long paths, as the number $n_{res} \sim r$ of "naive" resonances at $\varepsilon_p = E$ is proportional to the

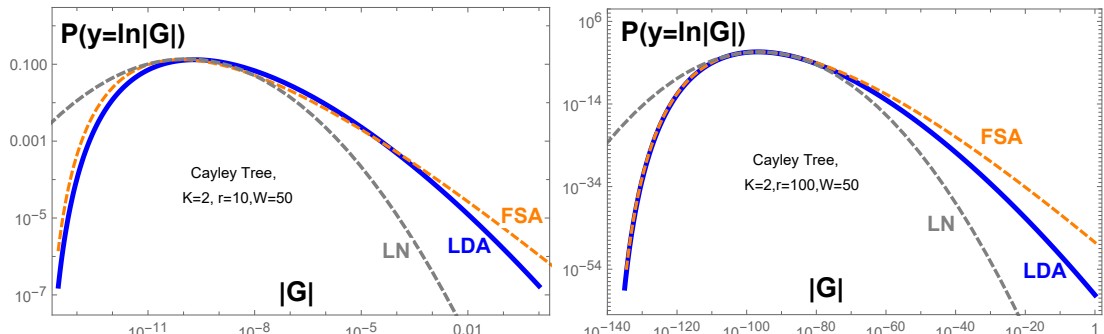

Figure 5: **Comparison of PDF of** $y = \ln|G|$ **on the** $K = 2$ **Cayley tree** for the large-$W$ approximation (LDA), Eq. (70) (solid blue line); the Poisson distribution in FSA, Eq. (2) (orange dashed line), and the log-normal (LN) (Gaussian in $\ln|G|$) distribution emerging from the Central Limit Theorem in FSA (gray dashed line) for two distances $r = 10$ (Left panel) and $r = 100$ (Right panel) at $W = 50$. The region $G_{typ} < |G| < 1$ of the solid blue curve corresponds to $0 < \beta < 1/2$, while the region $|G| < G_{typ}$ corresponds to the analytical continuation of $\epsilon_\beta$ to $\beta < 0$. The log-log derivative $d \ln P(y)/dy = -1$ at $|G| = 1$ which smoothly matches the tail $P(x = |G|) \sim G^{-2}$ at this point. The Poisson distribution emerging in FSA approximation at a finite $r$ overestimates the probability of large deviations $|G| > G_{typ}$ from the typical value $G_{typ}$ and the error increases with increasing the distance $r$. In contrast, in the region $|G| < G_{typ}$ the Poisson distribution becomes more accurate as $r$ increases. The log-normal distribution is valid at small and moderate deviations from the typical value where all three distributions nearly coincide.

length $r$. With self-energy parts taken into account, at most *one* resonance (out of $n_{res} \propto r$) may make uncompensated contribution to the product if it occurs at the last point of the path. Therefore the error of FSA increases with increasing $r$, as one can easily see comparing right and left panels of Fig. 5 and also in Fig. 1. This error depends only on the length of the path $r$ but not on the total length of a tree $L \gg r$.

Concluding this section, we would like to note that our "large-disorder" approximation for $\epsilon_\beta$ which is valid for $W \gg \sqrt{K-1}$, neglects the renormalization of the on-site disorder distribution, Eq. (15), and $\epsilon_\beta$ in this approximation does not depend on the branching number $K$. Thus it is also valid for strongly disordered one-dimensional Anderson model which corresponds to $K = 1$. This means that the distribution of the two-point Green's functions in this model is also described by Eqs. (79), (80), (81) with $\epsilon_\beta$ given by Eqs. (70)-(72). To the best of out knowledge this result for strongly disordered one-dimensional Anderson model is not known in the literature.

# 7 Conclusion and Discussion

Our goal was to investigate the applicability of the forward scattering approximation (FSA) and to evaluate the probability distribution function of the Green's function for the Anderson localization model on sparse graphs at large disorder. Such graphs represent the Hilbert space of interacting quantum systems and FSA is often suggested as the simplest tool to approach the problem of Many Body Localization (MBL). The Hilbert space of the realistic models of MBL include graphs like a hypercube lattice of Quantum Ising model and Quantum Random Energy model or its cross-section of XXZ Heisenberg chain. Such graphs have numerous loops and thus many shortest-length trajectories with correlated amplitudes that interfere and make

FSA rather complicated. However, even in the absence of such complication, on the Cayley tree, the applicability of FSA has severe limitations.

In this paper we have shown that FSA on the Cayley tree is not applicable in the limit of large distances $r$ between points in the two-point Green's function, however large (but fixed) disorder strength is. It strongly overestimates the probability of large deviations $|G_r| > G_{typ}$ from the typical value of Green's functions due to ignoring the correlated character of resonances along the path connecting the initial and the final points in a Green's function. Technically, this happens because FSA neglects the real parts of self energies of the single-point Green's functions which are responsible for the difference between the bare on-site energies $\varepsilon_p$ and the true eigenvalues $E_p$. The corresponding error increases with increasing the length of the path and in order to suppress it an unrealistically large disorder strength is required.

To arrive at this result, we computed the $r$-dependence of the moments of the real Green's function, relating them to the largest eigenvalue $\epsilon_\beta$ of the linearized transfer-matrix equation on a tree. This result, Eq. (3), is obtained by rigorous calculations in the framework of the Efetov's super-symmetry approach and it is valid at an arbitrary disorder strength. For strong disorder we derived a very accurate large-disorder approximations, Eqs. (70), (71), (72), for $\epsilon_\beta$ and checked its accuracy against a high-precision numerical solutions of the Abou-Chacra-Thouless-Anderson equations for the box distributed on-site disorder and by a comparison to exact solution of the problem for the Cauchy distribution.

Note that in the Anderson model on a two-dimensional lattice the FSA works very well [7] at strong disorder. Furthermore, the distribution function of $\ln(|G|/G_{typ}) = L^{1/3}\chi$ is broad with $\chi$ being well described by the Tracy-Widom distribution. It is neither of the form Eq. (79), nor it is Gaussian in $|G|$ in its central part, with non-Gaussian tails. This is only possible if the contributions of the different paths are strongly and non-trivially correlated to invalidate the Central Limit Theorem at all $|G|$. We believe that this is a peculiar property of a two-dimensional system which does not hold in other dimensions. Our results show that the distribution of $|G|$ is totally different in the one-dimensional Anderson model and there are indications [36] that it is strongly dimensionality-dependent. We believe that it is due to the strong dependence of the statistics of paths on dimensionality: 1D (as well as the Cayley tree) is special because of existence of a unique path, while $D = 2$ differs from $D = 3$ by statistics of loops. It is very important in the problem of weak localization that a random walker never returns to the origin in dimensionality $D > 2$ and does return for sure for a sufficiently long time in $D = 2$. It may be important for FSA too. Finally, the FSA may better work at higher dimensions $D > 1$ because the contribution of *multiple non-resonant* paths may dominate over that of a few resonant paths, thus making irrelevant the problem of a proper account for resonances.

We believe that our results, which emphasize the role of resonances, are encouraging to push forward the research on the distribution of Green's functions on the simplest realistic graphs emerging in the most popular models of MBL. The solution of this problem (to begin with the case of large disorder) would allow one to construct an equivalent random-matrix model of the Rosenzweig-Porter type [26, 27, 37] which is amendable to a number of approximate methods of the mean-field type.

In this respect we would like to mention a recent paper [38], where *broad distributions* of matrix elements responsible for *system-wide many-body resonances* have been numerically calculated in the many-body localized regimes of XXZ spin chains. These distributions are very similar to the one described by Eqs. (79), (80), (81), including the scaling with the system size $r \sim L/2$. Although the Hilbert space of these spin chains is a *hyper-cube* of a high dimension $L$ and the number of paths between the points at a distance $\sim L$ is very large $\sim (L/2)!$, the distribution of the matrix elements have fat tails which signals of the failure of the Central Limit Theorem. We attribute this to correlated contributions of the paths with different order of flipping the spins $s = 1/2$ which mutually cancel each other when close to resonance (see

e.g. [39]). This effect reduces the number of *statistically-independent* paths thus increasing the weight in the non-Gaussian tails making it possible for the moments of Green's functions of low order to be determined by the fat non-Gaussian tails, like in our example of a Cayley tree.

## Acknowledgements

We are grateful to B. L. Altshuler, M. V. Feigel'man and S. Raghu for insightful discussions.

**Funding information**   The work of P.A.N. was supported in part by the US Department of Energy, Office of Basic Energy Sciences, Division of Materials Sciences and Engineering, under contract number DE-AC02-76SF00515. V. E. K. acknowledges the Google Quantum Research Award "Ergodicity breaking in Quantum Many-Body Systems" and Abdus Salam ICTP for support during this work. In the part of numerical calculations, the work was supported by the Russian Science Foundation under the grant 18-12-00429 (A. K.). Analytical results were obtained with the support of the Russian Science Foundation under the grant 21-12-00409 (I. M. K.).

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
