# Peer review of "Statistics of Green's functions on a disordered Cayley tree and the validity of forward scattering approximation"

_SciPost Physics, doi:SciPost Phys. 12, 048 (2022)_

## Round 1 · Referee Report · Anonymous (Referee 1) · 2021-10-7

Strengths

  1. The comparison between an almost exact analytical method and an approximation widely used in quantum disordered systems, the forward approximation is interesting.

  2. The model considered is sufficiently simple and controlled so that the origin of the deviations of the forward approximation can be well understood.

  3. The paper contains quite non-trivial analytical developments which will allow future progress beyond comparison with the forward approximation.

Weaknesses

  1. The article is not, in its present form, easy to read. The analytical developments are too elliptical at some places.

  2. Much of the article consists of new analytical developments. But it is difficult to understand which are new, and which are already established in the literature.

  3. The only comparison with the forward approximation is finally a little disappointing. We would like, after these long and difficult calculations, to learn more than the mere fact that the forward approximation overestimates the effect of resonances.

Report

The article describes the problem of Anderson localization in a Cayley tree. This is an old problem, but which has been revisited recently with the highlighting of an interesting new property: the existence of a non-ergodic delocalized phase. This property is also found in a new class of random matrices that two of the authors recently discovered, and has been much debated recently regarding the Anderson transition in random graphs and many-body localization.
In this article, the authors develop an analytical approach based on supersymmetry, the basis of which was well established in this model. They are pushing this approach in an interesting new direction, the statistics of real Green's functions in the case of a finite Cayley tree. They then use their analytical predictions to examine to what extent the forward approximation, widely used in localized systems, describes the statistics of this observable. They find that the effects of resonances, the key mechanism that can lead to delocalization in this problem, are overestimated. Deviations are observed even at large disorder. This result is surprising because this approximation was supposed to be good in the regime of strong disorder, particularly in this tree where the loops are absent.
It is clear for me that this article deserves to be published in Scipost. However, I would like the authors to consider my following remarks and questions.

Requested changes

1.The article in its present form is not easy to read. The formalism used, supersymmetry, even if it is well established, in particular in the Cayley tree, would deserve to be described more precisely, in particular the links with the other formalisms used in this context. For example, section 2 is often too elliptical. a) It is important to remember what is the control parameter of the transition in equations (9-14). b) What does \Omega_0 represent physically? c) The Cayley tree does not always have a non-ergodic delocalized phase, it depends on the boundary conditions that we consider (the imaginary part of the energy E + i \eta at the boundary). In the population dynamics / pool method described in [26], it is shown that it is necessary to take eta-> N ^ {- \alpha}, with \alpha some exponent>0, to be able to observe this phase. If \eta is a constant (\alpha=0) on the contrary, we have an ergodic delocalized phase. How does this translate into the formalism presented? There is no discussion of the considered boundary condition. Are we forced to take Y (\Phi_n, \ Phi_n^\dagger) = 1? In the delocalized phase discussed briefly, the linearization of (9) is not always possible. What are the conditions for this linearization? d) The interpretation of (18) as a renormalized disorder strength is very interesting, however without a precise description of the physical meaning of the Lyapunov exponent \lambda_typ, it is very difficult to understand. Also, it would be useful to make the link with the more often used approach of the pool method / population dynamics. How does this renormalization of disorder appear? e) The computation of the moments of the real Green's function is done by means of the complex Green function (with the imaginary part \eta of E finite, and limit eta-> 0). We know that the limits N-> infty and eta-> 0 do not commute. I think the authors should stress that. f) The authors highlight an important difference between the model they consider and that corresponding to the non-linear sigma model, where an ergodic delocalized phase exists. Does this difference also change the nature of the non-ergodic localized / delocalized phase transition? g) The following calculations are more technical. Could the authors, before carrying them out, summarize the aspects which are new and those well known, and make a road map of these calculations?

  1. Concerning the forward approximation: a) It would be good in the introduction to recall the non-trivial successes of this approximation. For example, it makes it possible to understand that in dimension two the conductance in the Anderson localization regime has a Tracy-Widom distribution [PRL 99, 116602 (2007)] and glassy properties [PRL 122, 030401 (2019)] . b) My understanding is that this approximation neglects two main effects, which are related: the effect of loops (only directed paths, thus forward) and the non-linearity of the recursion due to the self-energies that we have not taken into account. From what I understand, the deviations observed with this approximation come here, in the Cayley tree, because the considered tree is finite. Because of this finitude, the contributing paths are not simply directed paths, but can go back and forth due to the presence of an edge. Or did I misunderstand, and even for an infinite tree, we have these corrections due to non-directed paths, which attenuate the effects of resonances? In all cases, the self-energies which appear due to these non directed paths, are only real in the case considered (real Green's functions, or \eta->0 before N->infty) and therefore only shift the resonances randomly. So is it their correlation that is important?

  2. The last remark I have relates to the fact that the analytical developments presented are very elaborate and that it is a little disappointing to conclude only on the forward approximation. I am convinced that other interesting predictions could be made. For example, it was recently shown that the Anderson transition in this type of graph of infinite effective dimensionality has two critical localization lengths [PR Research 2, 012020 (2020)]. Can this approach corroborate (or invalidate) these results for the Cayley tree? Can the authors give some perspectives for the future use of their analytical developments?

  • validity: good
  • significance: good
  • originality: good
  • clarity: ok
  • formatting: good
  • grammar: good

Author:  Ivan Khaymovich  on 2021-11-04  [id 1912]

(in reply to Report 1 on 2021-10-07)
Category:
answer to question

We are very grateful to the Referee for her/his highly professional review of our paper. Below we present the list of amendments resulting from the comments of the Referee.

  1. a) It is important to remember what is the control parameter of the transition in equations (9-14).

Reply: We thank the Referee for this clarifying question and answer to it on page 6 before Eq. (15) by the phrase: The Anderson transition “… is controlled by the energy $E$ and the disorder strength $W$ entering the characteristic function $\tilde{F}(v)$ of the distribution of on-site energies and thereby in $\epsilon_\beta$.”

b). What does $\Omega_0$ represent physically?

Reply: This is a very important question. We devoted an entire new Section 4 to answer it in a rigorous way. In addition, a short explanation about the physical meaning of this quantity was added on page 6 just after Eq. (15) and in the beginning of page 7. The discussion about the physical meaning of $\Omega_0$ was also added on page 5 around Eq. (11) and on page 6 after Eq. (13).

d) The interpretation of (18) as a renormalized disorder strength is very interesting, however without a precise description of the physical meaning of the Lyapunov exponent $\lambda_{typ}$, it is very difficult to understand. Also, it would be useful to make the link with the more often used approach of the pool method / population dynamics. How does this renormalization of disorder appear?

Reply: We added a paragraph at the end of Section 2 on page 7 explaining the physical meaning of the Lyapunov exponents in detail. In population dynamics numerics the Lyapunov exponents can be obtained by computing the two-point Green’s functions, as it was done, e.g., in Ref. [28].

e) The computation of the moments of the real Green's function is done by means of the complex Green function (with the imaginary part $\eta$ of $E$ finite, and limit $\eta \to 0$). We know that the limits $N\to \infty$ and $\eta\to 0$ do not commute. I think the authors should stress that.

Reply: We clarified this point on page 8 after Eq. (19). The limits $N\to \infty$ and $\eta\to 0$ do commute if the moment order $2\beta$ is less than $1$.

f) The authors highlight an important difference between the model they consider and that corresponding to the non-linear sigma model, where an ergodic delocalized phase exists. Does this difference also change the nature of the non-ergodic localized / delocalized phase transition?

Reply: At the end of Section 2 on page 7 of the revised manuscript we stressed that in the Anderson model with one orbital per site there is no the phase transition between non-ergodic and ergodic delocalized phases, in contrast to the non-linear sigma model. This was first demonstrated numerically by population dynamics method in Ref. [28] and shown analytically in Ref. [29] and in this paper. The reason is that the ergodic limit of the Lyapunov exponent is reached in the Anderson model with one orbital per site only in the limit of zero disorder.

g) The following calculations are more technical. Could the authors, before carrying them out, summarize the aspects which are new and those well known, and make a road map of these calculations?

Reply: This is very useful suggestion that helped us to improve the presentation of the results. We described such a roadmap in the beginning of Sec. 3 on page 7. All the results and methods of Sec. 3 are new.

c) The Cayley tree does not always have a non-ergodic delocalized phase, it depends on the boundary conditions that we consider (the imaginary part of the energy $E + i \eta$ at the boundary). In the population dynamics / pool method described in [26], it is shown that it is necessary to take $\eta\to N^{- \alpha}$, with $\alpha$ some exponent>0, to be able to observe this phase. If $\eta$ is a constant ($\alpha=0$) on the contrary, we have an ergodic delocalized phase. How does this translate into the formalism presented? There is no discussion of the considered boundary condition. Are we forced to take $Y (\Phi_n, \Phi_n^\dagger) = 1$? In the delocalized phase discussed briefly, the linearization of (9) is not always possible. What are the conditions for this linearization?

Reply: Here there is a certain misunderstanding. The point is that whether the phase is ergodic or not depends on the statistics of a single wave function. In order to distinguish between these two phases one has to have a resolution sufficient to speak about a single wave function. Such a resolution is only achievable if the broadening of an energy level $\eta$ is smaller than the level spacing $\delta=1/N$. We always imply the condition $\eta<1/N$ to be fulfilled considering the ‘real Green’s functions’. The boundary condition is indeed $Y = 1$ at the boundary which corresponds to the closed sample. However, we are mostly interested in the behavior far from the boundary considering $l<<R$. The discussion on the boundary condition and the order of limits was added on page 5 after Eq. (10).

2.a) It would be good in the introduction to recall the non-trivial successes of this approximation. For example, it makes it possible to understand that in dimension two the conductance in the Anderson localization regime has a Tracy-Widom distribution [PRL 99, 116602 (2007)] and glassy properties [PRL 122, 030401 (2019)] .

Reply: This is a correct comment. We added the reference to the above papers in the first sentence of the Introduction. We also discussed the first of these works in more detail in the Conclusion.

b) My understanding is that this approximation neglects two main effects, which are related: the effect of loops (only directed paths, thus forward) and the non-linearity of the recursion due to the self-energies that we have not taken into account. From what I understand, the deviations observed with this approximation come here, in the Cayley tree, because the considered tree is finite. Because of this finitude, the contributing paths are not simply directed paths, but can go back and forth due to the presence of an edge. Or did I misunderstand, and even for an infinite tree, we have these corrections due to non-directed paths, which attenuate the effects of resonances? In all cases, the self-energies which appear due to these non directed paths, are only real in the case considered (real Green's functions, or $\eta\to 0$ before $N\to\infty$) and therefore only shift the resonances randomly. So is it their correlation that is important?

Reply: The FSA, indeed, neglects the effect of loops considering only shortest paths. However, a different effect of loops, the existence of multiple paths between two points is generally taken into account within FSA. For the particular case of a tree, there are no loops in both senses. It is also true that the self-energies are not taken into account in FSA. However, it is not true that the effects of deviations from FSA on a tree is due to its finiteness. Even for an infinite tree and the real Green’s functions ($\eta \ll 1/N$)), the deviations will take place for a finite length r of a path between two points in the Green’s function. They are caused by the backscattering not from the boundary of a tree but rather because of the scattering by disorder. This backscattering causes correlations between the self-energies. We added a discussion on this point on pages 19 and 20.

  1. The last remark I have relates to the fact that the analytical developments presented are very elaborate and that it is a little disappointing to conclude only on the forward approximation. I am convinced that other interesting predictions could be made. For example, it was recently shown that the Anderson transition in this type of graph of infinite effective dimensionality has two critical localization lengths [PR Research 2, 012020 (2020)]. Can this approach corroborate (or invalidate) these results for the Cayley tree? Can the authors give some perspectives for the future use of their analytical developments?

Reply: We share the view of the Referee that our work opens up new perspectives of research. One of them is related to the distribution of Green’s functions on realistic graphs emerging in real interaction many-body systems. This discussion is added in the Conclusion. As for the existence of two characteristic lengths in the present problem or a similar problem on random regular graph, the answer is ‘yes’, we believe that the two length with the exponents 1 and ½ do exist in those problems. The discussion on this matter is present in our recent previous work Ref. [27], but we do not consider it relevant for the current manuscript.

---

## Round 2 · Referee Report · Anonymous (Referee 1) · 2021-11-11

Report

In the new version of their article, the authors have greatly clarified the points I had raised. I find the given physical interpretations very interesting, especially the new section 4.

I am not sure I agree with the statement in the conclusion that the agreement of numerical simulations (Refs. [6,7]) with the forward scattering approximation, in dimension two and in the strongly localized regime, is specific to this dimension. On the contrary, I believe that this agreement should persist in higher dimensions. My interpretation is that in the strongly localized regime, it is the competition between the paths that dominates (contrary to the weakly localized regime, where it is the interference between the paths which dominates). This competition is done by a global optimization, in a similar way to the physics of directed polymers. But I recognize that this remains an open problem which could be addressed in 3D or on random regular graphs for example. The question therefore arises as to whether the effects described by the authors are specific to 1D and to the Cayley tree where a single path connects two points of the network, or if these effects are important even in generic graphs where many paths contribute.

This discussion shows that the authors' interesting approach not only answers but also opens up interesting questions, in addition to being a technical `` tour de force ''. So I can only recommend the publication of the manuscript in SciPost.
  • validity: -
  • significance: -
  • originality: -
  • clarity: -
  • formatting: -
  • grammar: -

Author:  Ivan Khaymovich  on 2021-11-12  [id 1935]

(in reply to Report 1 on 2021-11-11)
Category:
remark

We thank the referee for his/her thorough reading of our revised manuscript and for high evaluation of changes.

We agree that the issue with FSA in higher dimensions in an open question, especially analytically.
We have only meant that in numerical simulations it seems that a specific Tracy-Widom form works only for 2D.

---

## Round 2 · Referee Report · Anonymous (Referee 2) · 2021-12-8

Strengths

1-New exact and approximate results on localisation physics in the Cayley tree, with potential applications to understanding both Anderson localisation and versions of many-body localization.

2-New results on the strongly disorder one-dimensional limit of Anderson localisation.

Weaknesses

Figures could be of higher quality

Report

Having read the paper and the reports from the first referee, I agree that this is a nice work with nontrivial results that deserve publication in SciPost Physics. I do not have much to add to the reports already written, though I have a couple of minor suggestions for improvements in the presentation that the authors can consider.

Requested changes

1-in the general discussion in the introduction, which considers all dimensions, the definition of r = |i-j| is not clear. 2-In Eq. (2) there is a P(y) on the left hand side, but no y on the right hand side. 3-All figures have rather clumsy looking labels that are placed in strange places. Since there are also legends put on the plots in the same style, it is not always obvious what is being plotted against what. I would suggest making the axis label in a more normal way. 4-The details of footnote 1, and the notation in the equations, could be more easily understood if the authors provided a figure defining the tree and the parameters r and R etc. This is maybe not needed for an expert, but it would be useful for those that are not experts on the method and trees but still interested in the results. 5-There are a couple of places where the wording could be improved. On page 6 between Eq. (14) and (15), they write "it appears to be at beta = 1/2". This "appears to be" makes the reader thing that it only appears to be at beta = 1/2 but is actually not there, but somewhere different. Since no explanation follows, I assume they mean to simply say that it is at beta = 1/2. After Eq. (19) the authors write "Upon averaging over disorder [...] the levels pass through the energy E just causing ...". This sentence is unclear. 6. $\theta$, $J_0$ and $K_{1-2\beta}$ are not defined when first introduced. One can guess these are Heaviside step function and Bessel functions, but it would be better to define them. 7. Eq. (42) is not an equation.

  • validity: top
  • significance: high
  • originality: high
  • clarity: good
  • formatting: reasonable
  • grammar: good

Author:  Ivan Khaymovich  on 2021-12-09  [id 2018]

(in reply to Report 2 on 2021-12-08)

The revised version is attached

Attachment:

Forward_v3_SciPost_IMK_VEK.pdf

Author:  Ivan Khaymovich  on 2021-12-08  [id 2016]

(in reply to Report 2 on 2021-12-08)

We are very grateful to the Referee for her/his highly professional review of our paper. Below we present the list of amendments resulting from the comments of the Referee.

1 - We thank the Referee for pointing this out. In the revised version we corrected $r = |i-j|$ to $r_{ij}$ throughout the text.

2 - We have corrected Eq. (2) and its discussion right beforehand by using the variable $z = \ln G + r \ln(W/2)$.

3 - We have modified the figures accordingly.

4 - We have added the corresponding figure and add some discussion into the text moved from the footnote.

5 - We have corrected the corresponding phrases.

6 - In the revised version of the manuscript, we have added the above definitions.

7 - As Sec. 3.3 is a technical derivation of the integration over variables $s_k$ and $\tilde s_k$, we take some freedom to use expressions, but not equations, like (39) and (42) where it is clear from reading the manuscript.

---

## Round 2 · Author Response

Dear Editor,

We are grateful to the referee for the high evaluation of our paper and for thorough careful reading of the manuscript. His/her critical remarks have allowed us to significantly improve the presentation of our work.

In the revised version of our manuscript, we address all the points mentioned by the referee.
The point-to-point reply to the referee is given below the report, while the list of changes is placed below.

Sincerely yours, the authors.

---

## Round 2 · List of Changes

1. On page 6 before Eq. (15) we have clarified the control parameter of the Anderson transition in Eqs. (9-14).
2. We have developed and described a physical meaning of the parameter $\Omega_0(v)$ on page 5 around Eq. (11), on page 6 after Eq. (13), on page 6 after Eq. (15), and in the beginning of page 7.
3. We have added a paragraph at the end of Section 2 on page 7 explaining the physical meaning of the Lyapunov exponents in the problem and the difference of a single-orbital model on the Cayley tree with respect to the non-linear sigma model.
4. We have described the roadmap of calculations in the beginning of Sec. 3 on page 7.
5. We have clarified the non-commutativity of limits of $\eta\to 0$ and $N\to\infty$ on page 8 after Eq. (19).
6. The discussion on the boundary condition and the order of limits has been added on page 5 after Eq. (10).
7. The discussion on the deviations of the forward scattering approximation from the exact analytical results of our paper has been added on pages 19 and 20.
8. The new perspective of research opened by our paper has been discussed in the Conclusion.
9. We have added the clarification of the meaning of the arguments t and v of $\Omega_r$ on page 5 before Eq. (11).
10. In order to avoid repeating notations, we have changed them in Eqs. (80-81).
11. Several relevant references (including the ones mentioned by the Referee) have been added to the Introduction and the Conclusion sections.
12. Several clarifications, minor amendments, and corrections of typos have been done throughout the manuscript.

---

## Editorial Decision

published